# Response Surface Methodology (RSM) Approach to Optimization of Coagulation-Flocculation of Aquaculture Wastewater Treatment Using Chitosan from Carapace of Giant Freshwater Prawn *Macrobrachium rosenbergii*

**DOI:** 10.3390/polym15041058

**Published:** 2023-02-20

**Authors:** Benedict Terkula Iber, Donald Torsabo, Che Engku Noramalina Che Engku Chik, Fachrul Wahab, Siti Rozaimah Sheikh Abdullah, Hassimi Abu Hassan, Nor Azman Kasan

**Affiliations:** 1Higher Institution Centre of Excellence (HICoE), Institute of Tropical Aquaculture and Fisheries (AKUATROP), Universiti Malaysia Terengganu, Kuala Nerus 21030, Malaysia; 2Department of Fisheries and Aquaculture, Joseph Sarwuan Tarka University (Formally Federal University of Agriculture, Makurdi), Makurdi P.M.B. 2373, Nigeria; 3Department of Chemical and Process Engineering, Faculty of Engineering and Built Environment, Universiti Kebangsaan Malaysia (UKM), Bandar Baru Bangi 43600, Malaysia

**Keywords:** chitin, chitosan, coagulation/flocculation, RSM, salinity, turbidity

## Abstract

The major sources of waste from aquaculture operations emanates from fish or shellfish processing and wastewater generation. A simple technique called coagulation/flocculation utilizes biowaste from aquaculture to produce chitosan coagulant for wastewater treatment. A chemical method was applied in the present study for chitin and chitosan extraction from carapace of *Macrobrachium rosenbergii* and subsequent application for removal of turbidity and salinity from shrimp aquaculture wastewater. Box-Behnken in RSM was used to determine the optimum operating conditions of chitosan dosage, pH, and settling time, after which quadratic models were developed and validated. Results show that 80 g of raw powder carapace yielded chitin and chitosan of 23.79% and 20.21%, respectively. The low moisture (0.38%) and ash (12.58%) content were an indication of good quality chitosan, while other properties such as water-binding capacity (WBC), fat-binding capacity (FBC), Fourier transform infrared spectroscopy (FTIR), X-ray diffraction (XRD), and scanning electron microscope (SEM) confirmed the structure and the α-group, as well as the rough morphology of chitosan. In addition, the high solubility (71.23%) and DDA (85.20%) suggested good coagulant potentials. It was recorded in this study that 87.67% turbidity was successfully removed at 20 mg/L of chitosan dosage and 6.25 pH after 30 min settling time, while 21.43% salinity was removed at 5 mg/L of chitosan dosage, 7.5 pH, and 30 min settling time. Therefore, the process conditions adopted in this study yielded chitosan of good quality, suitable as biopolymer coagulant for aquaculture wastewater treatment.

## 1. Introduction

The rapid growth and expansion in aquaculture have been attributed to the exponential increase in world population and a corresponding rise in global consumer demand [1]. As of 2016, fish production attained a global high peak of 171 million tones, where 47% was reported to have come from aquaculture [2]. Nevertheless, the impact of aquaculture wastewater such as environmental degradation and pollution of surface and underground water resources leading to occurrence of acute and chronic diseases have emanated from intensive aquaculture growth [3]. Wastewater from shrimp aquaculture, apart from being rich in nitrate and phosphate, is high in salinity, which makes biological treatment difficult [4]. In order to make good water available for the ever-increasing world population’s water consumption, as well as industrial development, wastewater from shrimp aquaculture should be treated before discharge or reuse [5]. One of the simple techniques of wastewater treatment is by coagulation/flocculation process, where a substance called coagulant is mixed with water to force the flocculation and settlement of the suspended particles [6].

The use of chemical coagulants in the coagulation/flocculation of suspended particles in various types of wastewaters has been widely reported by multiple sources [7,8,9,10]. However, the attendant negative effects of chemical coagulants, such as the production of high volumes of toxic sludge and precipitation of metal ions, have become a major concern to environmentalists, hence the introduction of biopolymer coagulants [10,11,12]. Biopolymer substances such as chitosan have been reported to possess the properties of both coagulant and flocculant. Apart from being an ecofriendly substance, it is nontoxic and easily biodegradable [13].

The production of chitin and chitosan has been reported from many sources, including crustaceans, mollusks, fungi, arthropods, scales, and some algae [14]. Specifically, chitin was successfully extracted from three mushrooms, namely: enoki (*Flammulina veltipe*), oyster (*Pleurotus ostreatus*), and shiitake (*Lentinula edodes*), where 23–35% chitin was obtained using mild alkaline extraction [15] The chemical extraction of chitin and chitosan usually begins with demineralization, which involves the removal of inorganic materials by dissolving the raw shell in acid, and it is followed by the elimination of proteins using alkaline solution in a process called deproteinization [16]. Many sources of chitin and chitosan contain pigments and are often eliminated by application of bleaching agents, and the pure chitin undergoes deacetylation in concentrated alkaline solution to produce chitosan [17,18]. It has been agreed by many researchers that for a deacetylated chitin to be called chitosan, the DDA must be above 50% but differ on the optimum conditions for the sequential steps in chitin and chitosan extraction [19,20,21,22]. One case is that of [23], who reported that 50 mL of 3% HCl, mixed at 1:10 *w*/*v* with 1 g of shrimp powdered shell, can successfully complete the demineralization process within one hour. In another development, chitosan was successfully explored from giant dung beetle, where HCl (1 M) was used at 95 °C for 2 h and 50% NaOH at the same temperature overnight to achieve demineralization and deacetylation, respectively [24]. Relatedly, the impact of different types of acid on the physicochemical properties of chitosan extracted from shrimp shell has also been investigated. Results show that H_2_SO_4_, HCl, and HNO_3_ were better in the removal of inorganic materials (demineralization) compared with organic acids [25].

Giant freshwater prawn, *Macrobrachium rosenbergii*, is a common shellfish species of the palaemonid freshwater prawn. It is mostly abundant in the tropical and subtropical areas of the Indo-pacific area. It is also available from India to southeast Asia and northern Australia. Many species of crustaceans have been used for extraction of chitin and chitosan [26,27]. However, there is limited literature on the isolation and characterization of chitosan from the carapace of giant freshwater prawn. It is believed that the hard and cutaneous carapace could be good sources of quality chitosan, just like many other crustaceans already reported. Chitosan has been applied in the coagulation/flocculation of wastewater from many sources, including pharmaceuticals, brewing, mining, sewage, and food processing [12,28,29,30]. The coagulation/flocculation process can be affected by a few factors, such as properties of the effluent water, pH, and dosage of the coagulant applied [31]. In order to achieve the best results of the process, the appropriate levels of these factors must be combined. A jar test apparatus is therefore recommended to determine optimum operating conditions of these factors [32]. However, the traditional method of optimization of the coagulation/flocculation process, which is by changing one interacting parameter while holding the other constant, is not only energy-consuming but also time-wasting. In addition, the method ignores the effect of the interaction of the operating factors; therefore, it cannot reveal the optimal combination of factors producing the desired responses [33].

In order to overcome the shortcomings of the traditional optimization process, a more robust design called Response Surface Methodology (RSM) is required [34]. RSM is a design capable of revealing interactions and the nonlinear dependencies of operating factors, thereby achieving a real optimum [35]. Therefore, this study was conducted to utilize chitosan isolated from the carapace of *M. rosenbergii* for removal of turbidity and salinity from real aquaculture wastewater. RSM was used for the optimization of the independent parameters (chitosan dosage, pH, and settling time) for greater removal efficiency.

## 2. Materials and Methods

### 2.1. Preparation and Characterization of Chitosan from Carapace of M. rosenbergii

#### 2.1.1. Collection and Preparation of Raw Material

The entire fresh samples were collected at the fish wet market in Terengganu, Malaysia. The dry carapace of *M. rosenbergii* (Figure 1) was removed. Samples were thoroughly washed to eliminate adhering particles, then evaporated in an oven at 70 °C until completely dry, before being ground using an electric blender to a particle mesh size of less than 20 nm.

#### 2.1.2. Chemical Preparation of Chitin and Chitosan

##### Demineralization

Demineralization was accomplished by mixing 80 g of dry powdered *M. rosenbergii* carapace with 1 M of hydrochloric acid (HCl) in a ratio of 1:10 (g/mL). The reaction was carried out at a temperature of 60 °C and spinning at 250 rpm for 2 h. After that, the sample was filtered and rinsed until the pH was neutral using running tap water. Samples were then further dried for 12 h at 70 °C in an oven.

##### Deproteinization

The dried, demineralized sample was deproteinized by adding 1 M NaOH at a solid/liquid ratio of 1:10 (g/mL) and a temperature of 100 °C. The reaction was carried out for two hours while being stirred at 250 rpm. The samples were filtered and pH-neutralized by running tap water washing. Samples were then further dried for 12 h at 70 °C in an oven.

##### Decoloration and Deacetylation

Chitin was submerged in 95% ethanol at a mass-to-volume ratio of 1:5 for 30 min at ambient temperature in order to enhance bleaching. Later, samples were cleaned and oven dried at 70 °C for 12 h. Subsequently, chitin was deacetylated by treating it with 60% NaOH at a solid/liquid ratio of 1:10 (g/mL) in a reaction vessel. For 2 h, the reaction was kept going at a temperature of 120 °C and an agitation speed of 250 rpm. The chitosan that resulted was filtered, extensively washed with running tap water until it reached a pH of 7, and then dried in an oven at 70 °C for 12 h.

#### 2.1.3. Characterization of Chitosan

In the present study, the properties of chitosan extracted from the carapace of *M. rosenbergii* were determined as expressed in Table 1.

### 2.2. Coagulation/Flocculation Using Chitosan from Carapace of M. rosenbergii

Wastewater samples were obtained from an intensive shrimp culture facility located in Bachok, Kelantan, Malaysia in plastic containers. Samples were then transferred to the water quality laboratory, Institute of Tropical Aquaculture and Fisheries (AKUATROP), Universiti Malaysia Terengganu, while held at 4 °C until used. Initial water quality parameters were tested and recorded as in Table 2 before coagulation/flocculation process.

#### 2.2.1. Preparation of Chitosan Coagulant Solution

Fresh stock solution of chitosan was prepared each day before beginning the coagulation/flocculation process. In total, 2 g of powder chitosan sample was mixed with 97 g of distilled water and 1 g of 1% acetic acid to create a favorable condition for the dissolution of chitosan particles [46,47]. The combination was further mixed thoroughly using a magnetic stirrer to ensure a homogenous solution.

#### 2.2.2. Experimental Design and Optimization

In this study, the incomplete factorial design (3^3^ of Box-Behnken design (BBD) in RSM) was used to optimize the process conditions for removal of turbidity and salinity from the shrimp aquaculture wastewater. The independent factors, which include chitosan dosage (mg/L), pH, and particles settling time (minutes), were coded as X_1_, X_2_, and X_3_, respectively, and varied at three levels (Table 3). The design of expert (DOE) in Statistica 12 software (Stöer Media, Berlin, Germany) was utilized to design the experiment and analyze obtained data. A total of 15 experimental runs were carried out and the outcomes recorded.

#### 2.2.3. Coagulation/Flocculation Process

Coagulation/flocculation of aquaculture wastewater was performed at the Water Quality Laboratory of the Institute of Tropical Aquaculture and Fisheries (AKUATROP), Universiti Malaysia Terengganu using the laboratory-scale jar test apparatus (JLT-6, Velp Scientifica, Usmate Velate, Italy) with 2 L square jars equipped with six-paddle stirrers. In this experiment, natural chitosan processed from the carapace of *M. rosenbergii* was utilized as coagulant. The removal percentages of turbidity (%) and salinity (%) were tested at room temperature.

Preliminary experiments were carried out to obtain the desired range of values for optimization. pH levels of the wastewater were adjusted by adding 0.1 M HCl or 0.1 M NaOH just before dosing of the coagulant. Fast and slow mixing of the sample and wastewater was achieved by means of automatic controller. The test apparatus was tuned at 150 rpm for 2 min for rapid mixing and 40 rpm for 30 min (slow mixing) to give room for flocculation. At the desired time of settling, the upper portion of the treated water was obtained and tested for turbidity and salinity.

#### 2.2.4. Analysis of Response Parameters

The percentage of turbidity removal by chitosan from the aquaculture wastewater was determined by the difference in the turbidity before and after treatment, expressed by 100 as in Equation (17). The initial and final turbidity of the wastewater were measured using a portable turbidity meter (HACH Model 2100P HACH Company, Loveland, CO, USA). Water salinity was measured using a refractometer (Vee Gee 43036 STX-3 Handheld, Lidköping, Sweden). Testing was carried out by placing a droplet of the water sample on a glass plate and sighting through the end to read a measurement of how far the light shining through the droplet is bent. This was repeated three times, and average values were taken. Percentage salinity removal was calculated using Equation (18).
(17)$$\mathrm{Turbidity}\;\mathrm{removal}\;\left(\%\right)=\frac{\mathrm{Initial}\;\mathrm{turbidity}-\mathrm{Final}\;\mathrm{turbidity}}{\mathrm{Initial}\;\mathrm{turbidity}}\times 100$$
(18)$$\mathrm{Salinity}\;\mathrm{removal}\;\left(\%\right)=\frac{\mathrm{Salinity}\;\mathrm{before}\;\mathrm{treatment}-\mathrm{Final}\;\mathrm{salinity}\;\mathrm{after}\;\mathrm{treatment}}{\mathrm{Salinity}\;\mathrm{before}\;\mathrm{treatment}}\times 100$$

#### 2.2.5. Fitness of Mathematical Model

The regression model designed for this study was tested for adequacy, where individual model coefficients and lack of fit were also tested for significance. In addition, statistical analysis of variance (ANOVA) was performed in order to assess the statistical significance as regards the fitness of the chosen quadratic model, as well as the significance of the individual response terms and their interactions. The general quadratic equation model is as shown in Equation (19).
(19)$$y_{i}=\beta^{0}+\overset{k}{\underset{i=1}{\sum }}\beta_{i}x_{i}+\overset{k}{\underset{i=1}{\sum }}{\beta_{ii}x_{i}}^{2}+\overset{k}{\underset{i<j}{\sum }}\beta_{ij}x_{i}x_{j}+\varepsilon$$
where *y_i_* represents the response, *x_i_* is the input factors, *β^0^*, *β_ii_* (*i* = 1, 2, …, *k*), _*ij*_ (*i* = 1, 2, … , *k*; *j* = 1, 2, …, *k*) are unknown parameters, and ε is a random error.

#### 2.2.6. Validation of Developed Model

In the present study, validity of the developed model was tested by conducting three separate runs of the experiment using the predicted values obtained during the optimization process for the removal of turbidity and color. Replicate values were analyzed using the independent student *t*-test at 95% confidence level [48].

## 3. Statistical Analysis

Using the SPSS statistical package application (SPSS 22.0. for Windows, SPSS Inc., Chicago, IL, USA), data from the extracted chitosan were analyzed. The arithmetic mean standard deviation was used to express the findings of batch tests that were carried out in triplicate. Using MiniTab and Statistica version 12, regression, ANOVA, and other analyses of the coagulation/flocculation experiment’s results were carried out.

## 4. Results and Discussion

The chemical extraction of chitosan from the carapace of *M. rosenbergii* resulted in the product displayed in Figure 2. Chitosan appeared pale white with a rough and sticky texture when felt.

### 4.1. Physicochemical Properties

The results of the physicochemical properties of chitosan extracted from the carapace of *M. rosenbergii* are presented in Table 4.

#### 4.1.1. Chitin and Chitosan Yield

In the present study, the process conditions established in the isolation of chitosan from the carapace of *M. rosenbergii* yielded 23.79% and 20.21% chitin and chitosan, respectively (Table 4). Studies have showed that the raw shell of crustaceans contains chitin, protein, calcium carbonate, phosphate, and other compounds [49,50]. To produce chitosan, inorganic elements were first removed by soaking raw shell in 1 M of HCl in a process called demineralization, followed by deproteinization with 1 M NaOH, and subsequently in 60% NaOH at high temperature to achieve deacetylation [51]. Results of chitin yield in this study are found to be higher than 4.3% from house cricket (*Brachytrupes portentosus*) [52], 5% of black soldier flies (*Hermetia illucens*) [53], 4.2% and 6.3% respectively ofcarapace and head of deep-sea mud shrimp [54], and 8.74% higher than freshwater crab (*Potamon algeriense*) [55]. However, a few other studies reported higher chitin yields from various sources using a similar technique applied in this study [56,57]. Similarly, the yield of chitosan after deacetylation compared favorably in this study with that from swimming crab (*Portunus trituberculatus*) [58] and *Penaeus vannamei* shrimp [59] and was found lower than that of others [28,43,56]. Therefore, the pale-white-color chitosan obtained from the present study can be said to have settled well with previous findings.

#### 4.1.2. Percentage Moisture and Ash Content

Moisture content is an important property of chitosan, as it is known to affect its physical properties and application. Chitosan easily absorbs moisture when exposed to air and is the reason why most commercial chitosan is stored in air-tight containers and kept at less than 10% moisture content [44]. In the present study, the 0.38% moisture content of chitosan from carapace (Table 4) is an indication of proper dryness and good quality. From [41]’s study using chitosan from shrimp shell biowaste, a moisture content 7.17% higher than the result of this study was reported. It has been reported that the level of ash content in chitosan is a reflection of the success or otherwise of the demineralization process [60]. Furthermore, high-purity chitosan is known to have an ash content of less than 1% [61]. Therefore, the less pure chitosan of 12.58% in this study may be a result of low acid concentration (1 M HCl) and shorter demineralization time (2 h). It is, however, worthy of mention that the ash content of chitosan from carapace was found to be lower than 13.62%, and 13.64% from shrimp chitosan [39].

#### 4.1.3. Fat-Binding Capacity (FBC) and Water-Binding Capacity (WBC)

The ability of chitosan particles to attract water molecules to themselves without dissolving in them is called WBC. This capacity has been demonstrated by chitosan from various sources and isolation techniques. For instance, studies have shown that chitosan from fish scales, crab, and shrimp possess WBCs of 492%, 138%, and 358% respectively [42]. In addition, 625.33% to 716.33% have also been reported from crayfish [38]. However, all these studies mentioned so far recorded WBCs higher than the 562.33% obtained in this study (Table 4). Nevertheless, in [62]’s study, it was submitted that the WBC of commercial chitosan ranges from 581 to 1150%, which is in agreement with the findings reported here. Results of FBC show the ability of extracted chitosan to bind with oil at 372.33%. This was found to be higher than the 217% to 403% obtained from crab legs by [21].

#### 4.1.4. Solubility and DDA

Although chitosan does not dissolve in water, it has been found to dissolve in 1% acetic acid. The level of dissolution of chitosan in this medium refers to its solubility. Solubility and DDA of chitosan are two important characteristics that determine its application, especially in the coagulation/flocculation process [23]. It has been established that several factors, such as the concentration of acid and alkali used during demineralization and deproteination, respectively, as well as the temperature and duration of heating, affect the solubility and DDA of chitosan [23,63]. This study observed that 60% NaOH applied in the deacetylation of chitin resulted in a solubility and DDA of 71.23% and 85.20%, respectively (Table 4). The solubility and DDA of chitosan in this study would have been higher than the values obtained but for the relatively high ash content [64]. It is on record that the solubility and DDA of chitosan obtained from insect cuticle [18], deep sea mud [54], and fungi [65] are all lower than the results of the present study.

#### 4.1.5. Bulk Density (BD), Tapped Density, Compressibility, Hausner Ratio (HR), and Carr’s Index (CI)

It is pertinent to state that the physical properties of chitosan, to a large extent, determine its area of application and are expressed by the shape and size of the chitosan particle. The shape and size of the chitosan particles determine its morphology and porosity [66]. The high bulk (0.25 g/mL) and tapped (0.32 g/mL) densities of chitosan from carapace are an indication of low porosity and are in agreement with 0.23 g/mL from crab and 0.21 g/mL from commercial chitosan [44]. The compressibility of chitosan, which indicates its relative difference in volume as pressure changes, was determined as 22.01. Furthermore, the ability of chitosan in the present study to aggregate and flow (CI), which was recorded as 21.02, can be considered as fair on the scale of flowability [67,68]. The relevance of the chitosan interparticle interaction, which was expressed in terms of compressibility and HR (1.27), was in agreement with the standard chitosan already in a previous report [69].

#### 4.1.6. Percentage of Inorganic Material, Protein, and Pigment in the Raw Sample

Results of the present study show that raw carapace shell of *M. rosenbergii* possesses 51.93% inorganic minerals, 18.70% protein, and 5.69% pigment. The separation of inorganic minerals and protein gives rise to chitin, while further deacetylation results in chitosan. In another study, inorganic minerals were reported at 40–55% and protein at 25–40% from shrimp shell [28] which were in agreement with the findings of the present study. Avelelas et al. [70] reported a 32.1% protein level in the shell of swimming crab, which was higher than that of crab carapace and 16.6% in the pereopods. Other studies reported 31.53% (crab) and 20.76% (shrimp) inorganic minerals [27], 12.5% minerals, and 38.7% protein in raw larvae of insect (*Hermetia illucens*) [71]. Generally, results of inorganic minerals and proteins in the present study are in tandem with previous findings from crustacean and other similar sources.

#### 4.1.7. Fourier Transform Infrared Spectroscopy (FTIR)

In order to verify the functional groups and further compare the isolated chitosan with commercial chitosan, FTIR was used. It was established in this study that chitosan from carapace showed various peaks in close semblance with the commercial one. The highest absorption peaks of primary amine and hydroxyl functional groups were observed at 3448 cm^−l^ and 3430 cm^−l^, respectively, for carapace (Figure 3a) and commercial chitosan (Figure 3b). It is evident in this study that intense bands of absorbance were found at the primary amine and the hydroxyl group. This highest absorption has also been reported by [26,72]. Abideen [42], and Bernabé [72] reported the peak for vibration of amine I (NH_2_) at 1624 cm^−1^, while 1654 cm^−l^ and 1654 cm^−l^ were observed in this study for commercial and carapace chitosan, respectively. Furthermore, amide III groups were located at 1383 cm^−l^ and 1383 cm^−l^ in commercial and carapace chitosan, respectively. Generally, the special absorption display around the NH indicates the occurrence of the deacetylation of chitin to chitosan [58]. Similarly, the asymmetric stretching of the C–O–C bridge was obtained from the commercial chitosan in the present study at 1073 cm^−1^. From the foregoing, it suffices to say that the chitosan isolated from carapace in this study falls under α- form [21].

#### 4.1.8. X-ray Diffraction (XRD)

The raw powder carapace of *M. rosenbergii* was put through XRD, and results show strong peaks at 20° and 30° (Figure 4a). The presence of chitin in the shell was revealed by the appearance of the peak at 20°, while that of 30° represents inorganic minerals [42,43]. Moreover, in this study, the diffraction intensity of carapace chitosan was the highest at 414 counts at 2θ values of 19.76° (Figure 4b) and compared favorably with commercial chitosan (230 counts at 19.86°) (Figure 4c). In addition, chitosan in the present study displayed two diffraction peaks at approximately 9° and 19°, which were higher than the peaks obtained from that of commercial ones, thereby confirming the more crystalline nature of the carapace chitosan. The disappearance of the strong reflection peak at 30° after processing of the raw shell was an indication of the success of the isolation process and final yield of chitosan [17,60]. Rasweefali et al. [54] reported a CrI of 33.8% and 39.6% from shrimp shell chitosan at different deacetylation temperatures, which were confirmed to be higher than those in this study (79.53%).

#### 4.1.9. Scanning Electron Microscope (SEM)

In the present study, chitosan from the carapace and commercial were tested for structural composition using SEM. The fine and irregular structures of carapace chitosan (Figure 5a) were a clear departure from that of commercial ones, which were found to be coarser in textural appearance (Figure 5b). The irregular appearance of carapace chitosan may be attributed to the effect of the chemical treatment of chitin and the drying process [43]. The smooth and less-fractured carapace chitosan in this study compares to that of [73] from insects.

### 4.2. Coagulation/Flocculation Using Chitosan from Dry Carapace of M. rosenbergii

The performance of chitosan extracted from the carapace of *M. rosenbergii* as coagulant was assessed through BBD, and the outcomes of the response parameters are presented in Table 5. The coded and levels of the independent variables (chitosan dosage, pH, and settling time) are also reported.

#### 4.2.1. Percentage Turbidity Removal Using Chitosan from Carapace of *M. rosenbergii*

The model developed for turbidity removal proved significant (model: *p* = 0.051). ANOVA results showed high R^2^ (88.62%) and R^2^ adjusted (68.13%) in addition to nonsignificant lack of fit (*p* = 0.149). Moreover, in the ANOVA result (Table 6), it can be observed that only the linear effect of settling time and the quadratic effect of pH were significant in the percentage removal of turbidity recorded. Furthermore, the model equation indicating both the significant and nonsignificant terms is presented in Equation (20).
polymers-15-01058-t006_Table 6Table 6ANOVA for response surface quadratic model on turbidity removal.SourceDFAdj SSAdj MSF-Value*p*-ValueModel970.71837.85764.330.051 Sig.Linear331.066910.35565.700.045 Sig.X_1_10.10580.10580.060.819X_2_111.021511.02156.070.057X_3_119.939619.939610.980.021 Sig.Square328.56959.52325.240.053 Sig.X_1_ × X_1_10.32310.32310.180.691X_2_ × X_2_126.190826.190814.420.013 Sig.X_3_ × X_3_11.24921.24920.690.4452-Way Interaction311.08193.69402.030.228X_1_ × X_2_15.83225.83223.210.133X_1_ × X_3_14.68724.68722.580.169X_2_ × X_3_10.56250.56250.310.602Error59.08201.8164

Lack-of-Fit38.15522.71845.870.149 Not Sig.Pure Error20.92690.4634

Total1479.8004


R^2^
88.62


R^2^ Adj.
68.13



% Turbidity removal = 0.9* + 0.719A + 24.28B + 0.002C* − 0.005A^2^ − 1.705B^2^* + 0.0037C^2^ − 0.129AB + 0.012AC − 0.024BC(20)
where X_1_ = A = dosage; X_2_ = B = pH; X_3_ = C = settling time.

Note: only terms with ***** in the equation are significant.

The optimization of turbidity removal using chitosan from the carapace of *M. rosenbergii* was set at 20 mg/L of chitosan dosage, 6.25 pH, and 30 min settling time (Figure 6a). These conditions were utilized to obtain 87.67% turbidity removal from the established model. It was further observed from the behavior of the desirability curves that settling time was still on the increase, indicating that more turbidity removal could be achieved at a higher settling time.

Figure 6b shows the pareto chart of turbidity removal using chitosan from the carapace of *M. rosenbergii*, where the bars are arranged in their order of significant effects. It is evident from the chart that the quadratic effect of pH was the most significant. The chart further corroborated the fact that, apart from the quadratic effect of pH and the linear effect of settling time, no other effect was found significant as far as turbidity removal using chitosan from the carapace of *M. rosenbergii* was concerned.

The 3D fitted response surface for turbidity removal is presented in Figure 6c. At constant chitosan dosage, the influence of turbidity removal can be expressed as displayed among the colors in the plot, where over 88% of turbidity could be removed. The darkest region is the surface where the effect is felt the most. The effect of pH was mostly felt within 6.2 to 7.0, while that of settling time was recorded within 28–30 min. It was further observed that pH values beyond 7.0 had a negative effect on turbidity removal, while settling time continued to rise.

In the present study, chitosan from the carapace of *M. rosenbergii* was used to carry out coagulation/flocculation for turbidity removal from aquaculture wastewater. The process variables were chitosan dosage, pH, and settling time. Results of the ANOVA from all the source show that the designed model was adequate for turbidity removal. For a model to be adequate, the coefficient of determination must be at least 80% and the lack of fit not significant in addition to good R^2^ adjusted [74,75]. The R^2^ values of 88.62% and lack of fit values of *p* = 0.148 proved that the model was adequate. This also suggested that the process variables and the levels of combination had significantly affected the removal of turbidity from the wastewater. Ghafari et al. [76] reported that for a model to be adequate, a high value of R^2^, which should be close to 100% and in agreement with the R^2^ adjusted, is desirable and necessary. The statement further posited that a satisfactory adjustment in the model data is ensured when the R^2^ is high. High R^2^ also shows that most of the total variation in the responses obtained were explained by the designed model [77].

It was recorded in this study that 87.67% turbidity removal was achieved at 20 mg/L chitosan dosage and pH of 6.25 after 30 min settling time. Due to the fact that the model designed for turbidity removal using carapace chitosan satisfied the condition for adequacy, and high removals were achieved, the developed quadratic model equation can be employed without the nonsignificant terms. To further demonstrate the level of significance of the process variables on turbidity removal, a pareto chart was plotted [78]. Results show that the quadratic effect of pH was the most significant factor influencing turbidity removal. The fitted response surface plot shows that less turbidity will be removed from the aquaculture wastewater at a higher pH above 7.0. The results of turbidity removal using chitosan from carapace of *M. rosenbergii* are higher than those of [79,80], who reported 74.5% and 46.84%, respectively, from wastewater.

#### 4.2.2. Percentage Salinity Removal Using Chitosan from Carapace of *M. rosenbergii*

The model for salinity removal was adjudged largely as significant considering the nonsignificant lack of fit (*p* = 0.058), which was strongly corroborated by high R^2^ (91.60%) and R^2^ adjusted (76.49%) (Table 7). It was observed that the only significant linear effect on salinity removal came from the dosage of chitosan applied. Nevertheless, the quadratic effects of pH and settling time, as well as interaction between dosage and settling, were also significant (*p* ≤ 0.05). Equation (21) represents the model equation for salinity removal using chitosan from the carapace of *M. rosenbergii*.
polymers-15-01058-t007_Table 7Table 7ANOVA for response surface quadratic model on salinity removal.SourceDFAdj SSAdj MSF-Value*p*-ValueModel9417.67246.4086.060.031 Sig.Linear389.30429.7683.890.089X_1_157.40657.4067.500.041 Sig.X_2_16.3726.3720.830.403X_3_125.52625.5263.330.127Square3213.45071.1509.290.017 Sig.X_1_ × X_1_12.9492.9490.390.562X_2_ × X_2_1144.173144.17318.830.007 Sig.X_3_ × X_3_173.65273.6529.620.027 Sig.2-Way Interaction3114.91838.3065.000.058X_1_ × X_2_112.74512.7451.660.253X_1_ × X_3_151.05151.0516.670.049 Sig.X_2_ × X_3_151.12251.1226.680.049 Sig.Error538.2887.658

Lack-of-Fit338.28812.7630.000.058 Not Sig.Pure Error20.0000.000

Total14455.961


R^2^
91.60


R^2^ Adj.
76.49



% Salinity removal = −136.7* + 1.10A* + 49.1B − 0.096C + 0.016A^2^ − 3.999B^2^* − 0.029C^2^* − 0.190AB − 0.038AC* + 0.229BC*(21)
where X_1_ = A = dosage; X_2_ = B = pH; X_3_ = C = settling time.

Note: only terms with ***** in the equation are significant.

Figure 7a shows the optimum conditions of the variable factors and the desirable percentage salinity removal. In total, 21.43% salinity removal was achieved by the application of 5 mg/L of chitosan dosage at 7.5 pH and 30 min settling time.

It is evident in Figure 7b (pareto chart) that the quadratic effect of pH was the most significant as far as percentage salinity removal using chitosan from the carapace of *M. rosenbergii* was concerned. In their order of significance, the next significant effect was that of settling time (quadratic) before the linear effect of chitosan dosage. As usual, only t-statistical values on the right-hand side of the vertical red line were considered as significant at a 95% confidence level.

The fitted surface for percentage salinity removal (Figure 7c) shows that the highest response area of the most significant process variables is located at the darkest region of the plot. Over 24% salinity can be achieved through the effect of pH and settling time at constant chitosan dosage. It was evident that salinity removal rose with an increase in the variable factors to a certain level and then declined.

Many aquaculture systems are carried out using marine water and, as such, the wastewater generated needs to be treated before discharge into the environment. High salinity aquaculture wastewater has been reported to adversely affect adjacent lands and other water bodies, including mangrove wetlands [80]. Although many studies have expressed the difficulty in salinity removal using the coagulation/flocculation process [81,82], this study recorded a little success in this regard. With an R^2^ of 91.60% and R^2^ adjusted of 76.49, as well as a nonsignificant lack of fit value of 0.058, the model terms were considered to have exerted significant effect on salinity removal from the aquaculture wastewater. It was recorded that 21.43% salinity removal was achieved using 5 mg/L of chitosan dosage at 7.5 pH and 30 min settling time. This result is higher than the 15% reported by [53]. Therefore, the developed equation for the model can be applied without the nonsignificant terms in order to eliminate a reasonable quantity of dissolved salts and lower the salinity of the wastewater before discharge. The Pareto chart which ranks the effects of the independent variables [83] shows that the quadratic effect of pH was the most significant in salinity removal. In addition, fitted surface plots generated by the Statistica 12 software described the behaviors in the salinity removal as the most significant variables. This revealed that salinity removal rose with an increase in the variable factors to a certain level and then declined.

#### 4.2.3. Validation of the Adequacy of Models for Turbidity and Salinity Removals

The adequacy of the designed model was validated in a separate experiment. For each parameter, optimized conditions were tested in three replicate experiments, and the outcomes, which were analyzed using one-sample *t*-test, are presented in Table 8. In all cases, there was no significant difference (*p* ≥ 0.05) between the actual and predicted values.

## 5. Conclusions

Chitosan was isolated through the removal of inorganic minerals (demineralization using 1 M HCl), proteins (deproteinization using 1 M NaOH), pigments (decoloration using 95% ethanol), and acetyl groups (deacetylation using 60% NaOH) from the carapace of *M. rosenbergii*. Chitin and chitosan yields were 23.79% and 20.21%, respectively, while 0.38% moisture content was an indication of good quality chitosan. The high bulk and tapped densities of the isolated chitosan suggested low porosity, while the CI value showed chitosan was of the fair flowability category. Similarly, the high solubility (71.23%) and DDA (85.20%) reported from the carapace chitosan is an indication of a good coagulant. The process conditions employed in the present study were effective in eliminating the minerals and proteins found in the raw carapace of *M. rosenbergii*. The high chitin yield was an indication that the source is an important choice for the production of the biopolymer. Chitosan properties such as ash content, WBC, FBC, FTIR, XRD, and SEM compared favorably with reported findings and those of commercial chitosan. The various physicochemical and morphological properties evaluated in this study will assist end users in relating the chitosan with applications of interest. It is, however, difficult to conclude the exact optimal ranges of the biopolymer parameters, since these properties vary with usage or application.

This study determined that the models designed for turbidity and salinity removals were adequate; having R^2^ (88.62%), R^2^ adjusted (68.13%), and a lack of fit value (*p* = 0.1419) for turbidity removal and R^2^, R^2^ adjusted, and a lack of fit value of 91.60%, 76.49%, and *p* = 0.058, respectively, for salinity removal. Chitosan from carapace successfully removed 87.67% turbidity at 20 mg/L dosage and 6.25 pH after 30 min settling time, while 21.43% salinity removal was achieved at 5 mg/L of chitosan dosage at 7.5 pH and 30 min settling time. Finally, validation of the optimized process conditions as suggested by the quadratic model showed that the model predictions were accurate. In the study, the limitations of the classical methods of factor combination were overcome using RSM. This was adequately demonstrated by showing where the optimum operating conditions lied, while at the same time displaying the effects of their interactions on the removal of turbidity and salinity. From the behavior of the independent variables, it was apparent that more turbidity removal could be achieved at higher settling time. Overall, it can be concluded that chitosan from *M. rosenbergii* can be used for the pretreatment of aquaculture wastewater as a substitute for chemical coagulant. It is recommended that further studies be conducted on other sources of wastewater.

## Figures and Tables

**Figure 1 polymers-15-01058-f001:**
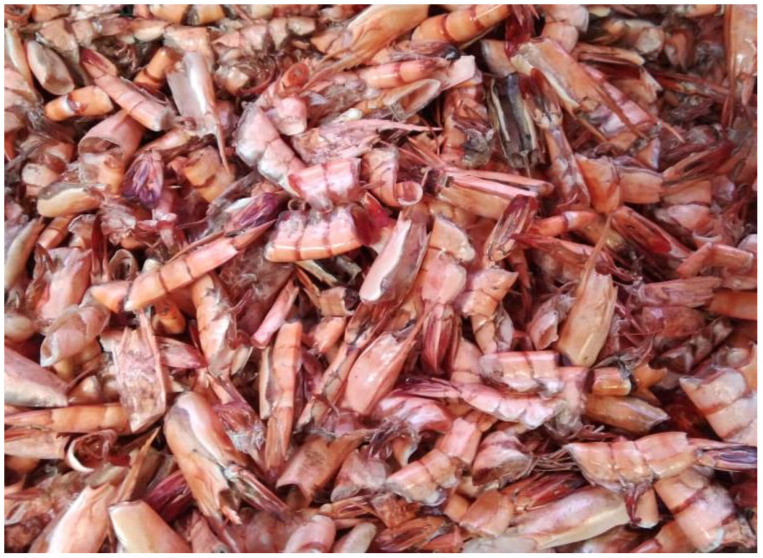
Dry carapace of *M. rosenbergii*.

**Figure 2 polymers-15-01058-f002:**
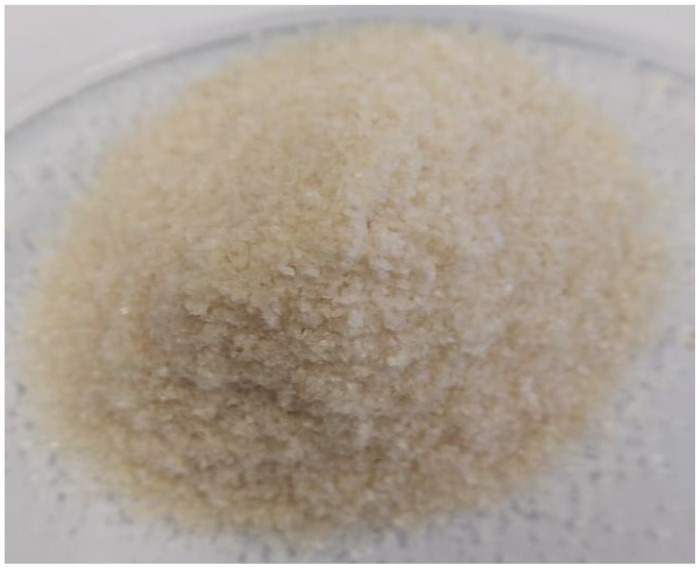
Chitosan from dry carapace of *M. rosenbergii*.

**Figure 3 polymers-15-01058-f003:**
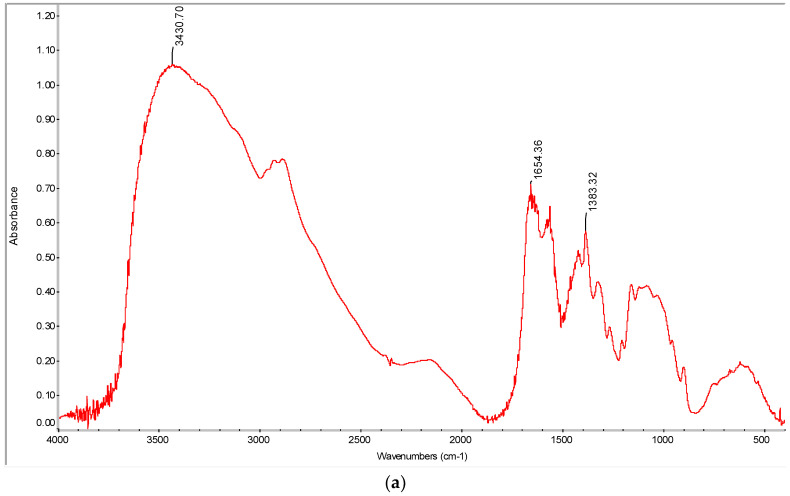
(**a**) FTIR spectra of chitosan from carapace of *M. rosenbergii*; (**b**) FTIR spectra of commercial chitosan.

**Figure 4 polymers-15-01058-f004:**
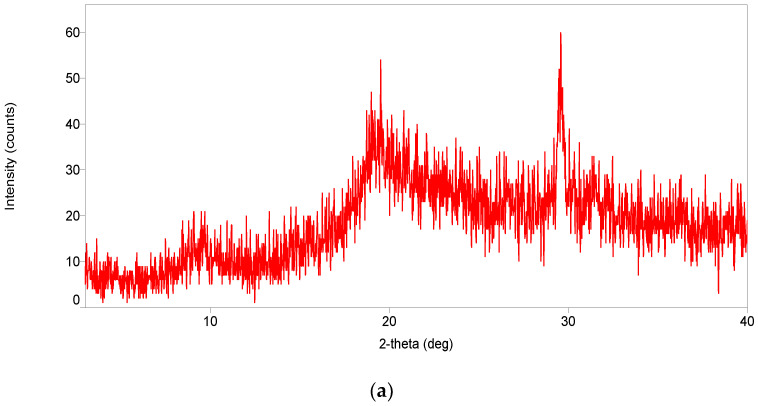
(**a**) XRD spectra of raw dry carapace powder of *M. rosenbergii*. (**b**) XRD spectra of chitosan from carapace of *M. rosenbergii.* (**c**) XRD spectra of commercial chitosan.

**Figure 5 polymers-15-01058-f005:**
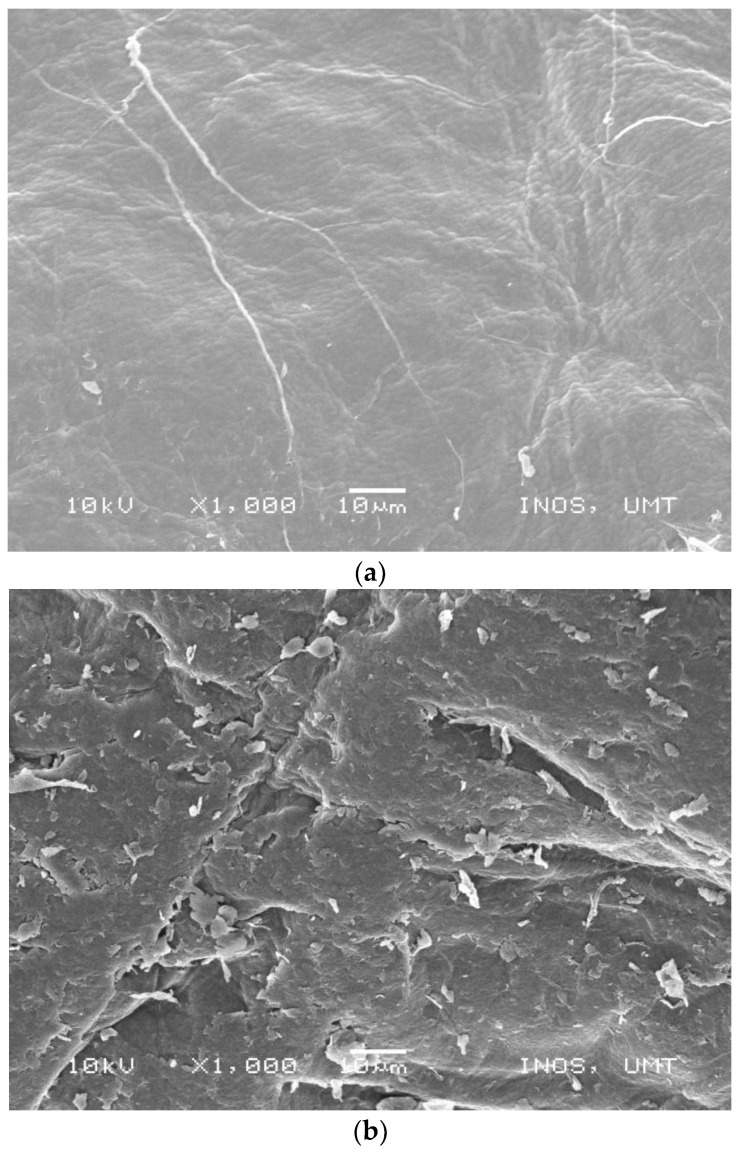
(**a**) SEM image of chitosan from carapace of *M. rosenbergii*. (**b**) SEM images of commercial chitosan.

**Figure 6 polymers-15-01058-f006:**
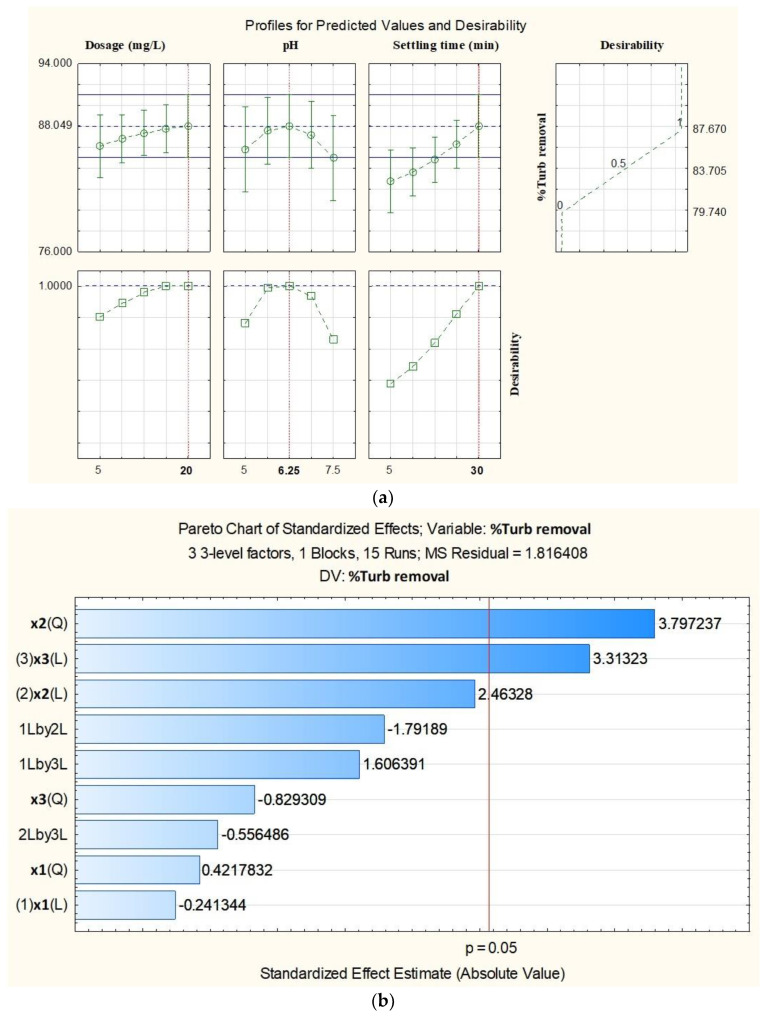
(**a**) Predicted and desirable % turbidity removal using chitosan from carapace of *M. rosenbergii.* (**b**) Pareto chart for standardized effects of the independent factors and interactions for % turbidity removal using chitosan from carapace of *M. rosenbergii*. (**c**) Three-dimensional fitted surface plots of % turbidity removal using chitosan from carapace of *M. rosenbergii*.

**Figure 7 polymers-15-01058-f007:**
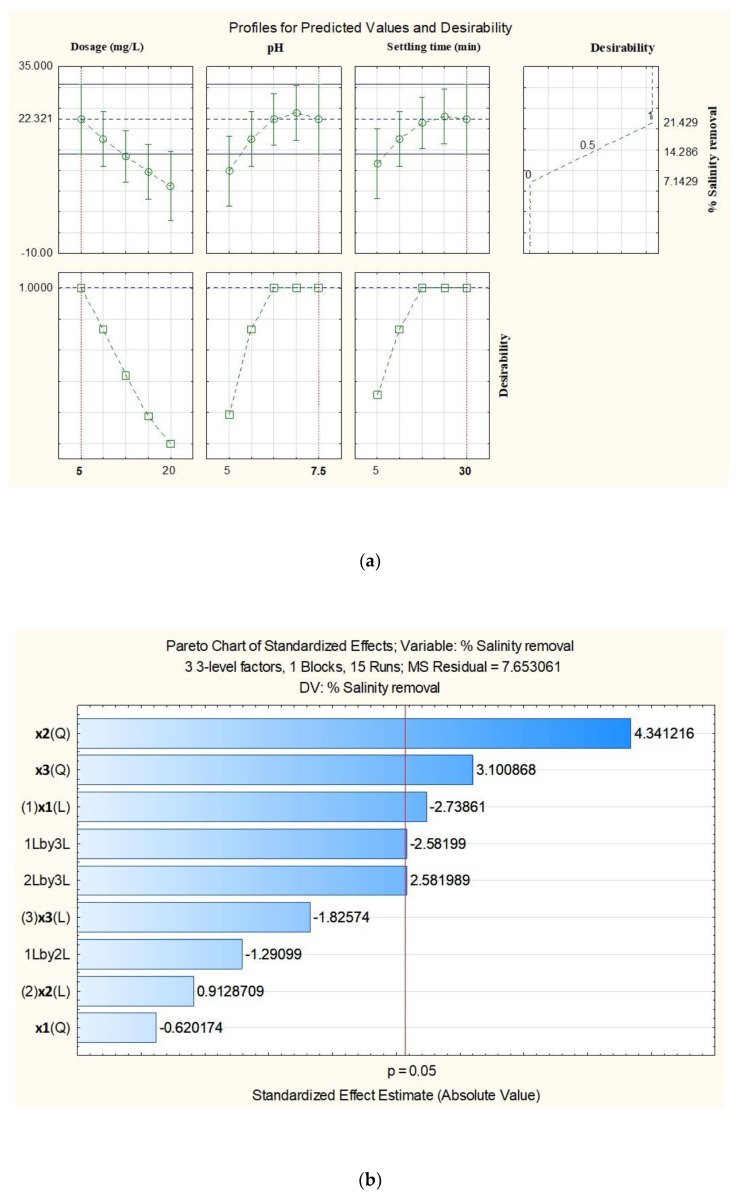
(**a**) Predicted and desirable % salinity removal using chitosan from carapace of *M. rosenbergii.* (**b**) Pareto chart for standardized effects of the independent factors and interactions for % salinity removal using chitosan from carapace of *M. rosenbergii.* (**c**) Three-dimensional fitted surface plots of % salinity removal using chitosan from carapace of *M. rosenbergii*.

**Table 1 polymers-15-01058-t001:** Characterization of chitosan from the carapace of *M. rosenbergii.* Adapted from [36].

Parameter	Method	Equation	
**Chitin and chitosan yield**	Chitosan is the weight that remains after chitin has been deacetylated, whereas chitin was measured by comparing weight measurements taken before and after mineral elements and protein were removed from the raw powder [37].	$\mathrm{Chitin}\;\mathrm{yield}\;\left(\%\right)=\frac{\mathrm{Extracted}\;\mathrm{chitin}\;\left(g\right)}{\mathrm{Ground}\;\mathrm{shell}\;\mathrm{of}\;\mathrm{shrimp}/\mathrm{prawn}\;\left(g\right)}\times 100$	(1)
		$\mathrm{Chitosan}\;\mathrm{yield}\;\left(\%\right)=\frac{\mathrm{Extracted}\;\mathrm{chitosan}\left(g\right)}{\mathrm{Extracted}\;\mathrm{chitin}\;\mathrm{of}\;\mathrm{shrimp}/\mathrm{prawn}\;\left(g\right)}\times 100$	(2)
**Percentage moisture**	In [37,38] was used the gravimetric method, which makes use of a hot air oven, to determine the moisture content of chitosan. The samples were heated to a consistent weight in an oven at 70 °C. By comparing the differences in weight before and after drying, moisture content was determined.	Moisture (%)$=\frac{\mathrm{Wet}\;\mathrm{weight}\;\mathrm{of}\;\mathrm{the}\;\mathrm{sample}\;\left(g\right)-\mathrm{Dry}\;\mathrm{weight}\;\mathrm{of}\;\mathrm{the}\;\mathrm{sample}\;\left(g\right)}{\mathrm{Wet}\;\mathrm{weight}\;\mathrm{of}\;\mathrm{the}\;\mathrm{sample}\;\left(g\right)}\times 100$	(3)
**Ash content**	One gram of the sample was put in a silica crucible and roasted to 600 °C for five hours in a muffle furnace to determine the sample’s ash concentration. Samples were placed in desiccators after being further cooled to 200 °C. Ash content was calculated using the weights of the crucible and the ash leftover.	$\mathrm{Ash}\;\mathrm{content}\;\left(\%\right)=\frac{\mathrm{Weight}\;\mathrm{of}\;\mathrm{the}\;\mathrm{ash}\;\mathrm{residue}\;\left(g\right)}{\mathrm{Sample}\;\mathrm{weight}\;\left(g\right)}\times 100$	(4)
**WBC**	WBC was measured using the technique described by [39]. A centrifuge tube was filled with 0.5 g of chitosan sample before 10 mL of distilled water was mixed in. To dissolve the chitosan, the liquid was then vortexed for one minute and left at room temperature for 30 min. The tube was then centrifuged for 25 min at 3200 rpm after shaking for 5 s every 10 min. The tube was weighed again to determine the water bound after decantation of the supernatant.	$\mathrm{WBC}\;\left(\%\right)=\frac{\mathrm{Water}\;\mathrm{bound}\;\left(g\right)}{\mathrm{Initial}\;\mathrm{sample}\;\mathrm{weight}\;\left(g\right)}\times 100$	(5)
**FBC**	Refs. [40,41] assessed the chitosan’s fat-binding capacity (FBC) using a modified method. In order to measure FBC, a centrifuge tube containing 0.5 g of chitosan sample, 10 mL of soybean oil, and 1 min of vortex mixing to disperse the samples was gauged. The mixture was centrifuged at 3000 rpm for 25 min after being held at room temperature for 30 min and shaking for 5 s every 10 min. The supernatant was then drained, and the cylinder was reweighed after that.	$\mathrm{FBC}\;\left(\%\right)=\frac{\mathrm{Fat}\;\mathrm{bound}\left(g\right)}{\mathrm{Initial}\;\mathrm{sample}\;\mathrm{weight}\;\left(g\right)}\times 100$	(6)
**Solubility**	Chitosan’s solubility in weak acidic medium was assessed using a modified version of [37,42]’s methods. To create 1% chitosan solution, one gram of chitosan was treated with 1% acetic acid solution. This solution was swirled with a magnetic stirrer at ambient temperature for two hours. The mixture was then centrifuged at 600 rmp for five minutes, and then filtered through Whatman No. 1 filter paper that was preweighed (Wi). The filter paper was reweighed after being further dried at room temperature (Wf).	$\mathrm{Solubility}\;\left(\%\right)=100-\left[\frac{wf-wi}{Ws}\times 100\right]$	(7)
		where *Wi* and *Wf* refer to the initial and final weight of filter paper, while *Ws* is the weight of substance (chitosan)	
**DDA**	For FTIR spectra analysis utilizing an I.R. instrument, chitosan samples were made in KBr disks and film (MB- 100, Bomem Hartmann & Braun, Quebec, Canada). Following frequency set to 4000–400 cm^−1^, DDA was calculated using the technique suggested by [42].	DDA (%) = $100-\left[\left(\frac{A1655}{3450}\right)\times \frac{100}{1.33}\right]$	(8)
		where DDA is the degree of deacetylation; A1655 is the peak area for the band at 1655 cm^−1^; A3450 is the peak area for the band at 3450 cm^−1^; and 1.33 is the factor representing the ratio of A1655/A3450 for complete N-acetylated chitosan.	
**Bulk density (BD)**	According to [43]’s study, the bulk density (BD) of the chitosan samples can be estimated as a function of the mass and volume occupied by the given sample. A chitosan sample weighing 5 grams was put into graduated centrifuge tube, and volume was recorded without shaking. To determine an average volume, this process was performed five times.	$\mathrm{Bulk}\;\mathrm{density}\;\left(g/\mathrm{mL}\right)=\frac{\mathrm{Mass}\;\mathrm{of}\;\mathrm{the}\;\mathrm{sample}}{V}$	(9)
		where V is the untapped volume of sample in the centrifuge tube	
**Tapped density (TD)**	A chitosan dry sample weighing 5 g was inserted in a calibrated centrifuge tube and mixed thoroughly until a consistent volume was achieved in order to determine the tapped density of the material. For all samples, the experiment was run 3 times.	$\mathrm{Tap}\;\mathrm{density}\;\left(g/\mathrm{mL}\right)=\frac{\mathrm{Mass}\;\mathrm{of}\;\mathrm{the}\;\mathrm{sample}}{\mathrm{Vtap}}$	(10)
		where Vtap is the volume of the substance in the centrifuge tube after tapping or shaking	
**Compressibility**	In this investigation, the proportional variation in the volume of the substance in reaction to pressure or a change in mean stress was used to estimate the compressibility of dry powder chitosan.	$\mathrm{Compressibility}=\frac{100\left(\mathrm{Vo}-\mathrm{Vf}\right)}{\mathrm{Vo}}$	(11)
		where V_o_ is the unsettled apparent volume, while V_f_ is the final volume after tapping.	
**Hausner ratio (HR)**	The frictional tensions between the granules of chitosan are shown by the HR of samples.	$\mathrm{Hausner}\;\mathrm{ratio}\;\left(\mathrm{HR}\right)=\frac{\mathrm{Dtap}}{\mathrm{Dbulk}}$	(12)
		where Dtap and Dbulk are the tapped and bulk densities of the chitosan samples, respectively	
**Carr’s index (CI)**	CI stands for cohesion index and describes the capacity of the chitosan particles to aggregate.	${\mathrm{Carr}}^{’}s\;\mathrm{index}\;\left(\mathrm{CI}\right)=\frac{\mathrm{Dtap}-\mathrm{Dbulk}}{\mathrm{Dtap}}\times 100$	(13)
		where Dtap and Dbulk are the tap and bulk densities of the substance, respectively	
**% Inorganic**		$\mathrm{Inorganic}\;\mathrm{removal}\;\left(\%\right)=\frac{\mathrm{Mass}\;\mathrm{before}\;\mathrm{DM}-\mathrm{Mass}\;\mathrm{after}\;\mathrm{DM}}{\mathrm{Mass}\;\mathrm{before}\;\mathrm{DM}}\times 100$	(14)
		where DM: demineralization	
**% Protein**		$\mathrm{Protein}\;\mathrm{removal}\;\left(\%\right)=\frac{\mathrm{Mass}\;\mathrm{before}\;\mathrm{DP}-\mathrm{Mass}\;\mathrm{after}\;\mathrm{DP}}{\mathrm{Mass}\;\mathrm{after}\;\mathrm{DP}\;}\times 100$	(15)
		where DP: deproteinization	
**% Pigment**		$\mathrm{Pigment}\;\mathrm{removal}\;\left(\%\right)=\frac{\mathrm{Mass}\;\mathrm{before}\;\mathrm{DC}-\mathrm{Mass}\;\mathrm{after}\;\mathrm{DC}}{\mathrm{Mass}\;\mathrm{before}\;\mathrm{DC}}\times$100	(16)
		where DC: decoloration	
**X-ray diffraction**	To determine the crystalline nature of the chitosan, wide-angle X-ray diffraction investigations were performed using a diffractometer XRD (Bruker model D8 ADVANCE), operated at a voltage of 40 V and a current of 30 mA with Cu K radiation (=1.54060). The XRD pattern was captured in a fixed-time mode at ambient temperature in the 2θ range of 9 to 80 degrees [44].		
**SEM**	Using a scanning electron microscope (JEOL, JSM-7600 F, Japan), morphological characterization of the granular chitosan surface was carried out at 1000× magnification. Therefore, Chitosan samples’ dimensions, forms, and shapes were examined [45].		
**FTIR**	After samples were made in KBr disks and film, infrared spectra of the chitosan samples were acquired using I.R. equipment (MB-100, Bomem Hartmann & Braun, QC, Canada). The range of frequency was 4000–400 cm^−1^ [46].		

**Table 2 polymers-15-01058-t002:** Initial water quality parameters of aquaculture wastewater from Bachok farm, Kelantan.

Parameter	Unit	Value	Environmental Standard
Temperature	°C	23.82 ± 1.15	40
Dissolved oxygen	mg/L	2.8 ± 2.12	>3
Salinity	ppt	13.21 ± 0.44	
Turbidity	NTU	81 ± 2.22	<0.15
Total suspended solid	mg/L	86 ± 3.51	50–100
pH	-	7.95 ± 3.11	6.0–9.0
Nitrite (NO_2)_	mg/L	1.50 ± 34	1
Ammonia (NH_3_)	mg/L	0.86 ± 0.55	0.25
Phosphate (PO_4_)	mg/L	11.37 ± 1.03	0.05

**Table 3 polymers-15-01058-t003:** Experimental ranges of factors examined using Box–Behnken design for monodon.

Variables	Symbol	Levels of Variation		
		−1	0	+1
pH	X1	5.00	6.25	7.50
Coagulant dosage (mg/L)	X2	5.00	12.50	20.00
Settling time (min)	X3	5.00	17.50	30.00

**Table 4 polymers-15-01058-t004:** Physicochemical parameters of chitosan.

Parameters	Mean Values
Chitin yield (g)	11.98 ± 0.49
Percentage chitin yield (%)	23.79 ± 1.12
Chitosan yield (g)	10.10 ± 0.02
Percentage chitosan yield (%)	20.21 ± 0.23
Percentage moisture content (%)	0.38 ± 0.13
Ash (%)	12.58 ± 0.51
WBC (%)	562.33 ± 7.51
FBC (%)	372.33 ± 3.51
Solubility (%)	71.23 ± 7.64
Bulk density (g/mL)	0.25 ± 0.44
Tapped density (g/mL)	0.32 ± 0.06
Compressibility	22.01 ± 1.94
Hausner ratio	1.27 ± 0.05
Carr’s index	21.02 ± 6.14
Percentage inorganic material (%)	51.93 ± 4.89
Percentage protein (%)	18.70 ± 8.26
Percentage pigment (%)	5.69 ± 3.44
DDA (%)	85.20 ± 4.49
Color	Pale white

Values are presented as mean ± standard deviation.

**Table 5 polymers-15-01058-t005:** Design matrix and BBD experimental results of turbidity and salinity removal using chitosan from carapace of *M. rosenbergii*.

Runs	x1	x2	x3	Dosage (mg/L)	pH	Settling Time (Min)	%Turb Removal	%Salinity Removal
1	−1	−1	0	5	5	17.5	79.88	14.29
2	1	−1	0	20	5	17.5	81.27	14.29
3	−1	1	0	5	7.5	17.5	85.66	21.43
4	1	1	0	20	7.5	17.5	82.22	14.29
5	−1	0	−1	5	6.25	5	85.5	21.43
6	1	0	−1	20	6.25	5	83.9	21.43
7	−1	0	1	5	6.25	30	84.94	21.43
8	1	0	1	20	6.25	30	87.67	7.14
9	0	−1	−1	12.5	5	5	79.74	14.29
10	0	1	−1	12.5	7.5	5	81.82	7.14
11	0	−1	1	12.5	5	30	85.2	7.14
12	0	1	1	12.5	7.5	30	85.78	14.29
13	0	0	0	12.5	6.25	17.5	84.44	21.43
14	0	0	0	12.5	6.25	17.5	85.71	21.43
15	0	0	0	12.5	6.25	17.5	85.5	21.43

**Table 8 polymers-15-01058-t008:** Validation test for the predicted response parameters from carapace of *M. rosenbergii* chitosan.

Parameter	Predicted	Actual	Difference
% Turbidity removal	87.67	90.80 ± 0.35	3.13
% Salinity removal	21.43	22.40 ± 1.80	0.97

No statistically significant difference between predicted and actual (*p* ≤ 0.05).

## Data Availability

The data presented in this study are available on request from the corresponding author.

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
