# Peer review of "Response Surface Methodology (RSM) Approach to Optimization of Coagulation-Flocculation of Aquaculture Wastewater Treatment Using Chitosan from Carapace of Giant Freshwater Prawn Macrobrachium rosenbergii"

_polymers, 2023, doi:10.3390/polym15041058_

Round 1

Reviewer 1 Report

This paper describes a simple coagulation/flocculation technique that uses aquaculture biowaste to produce chitosan coagulants for the removal of turbidity and salinity from actual aquaculture wastewater. Response Surface Methodology (RSM) was used to optimize chitosan dosage, pH, settling time and other independent parameters to obtain a higher removal efficiency. The research idea is interesting and the potential application is valuable. I believe that this paper has an important contribution to the use of RSM approach to optimize new applications of Macrobrachium rosenbergii crustaceans for the coagulation-flocculation treatment of aquaculture wastewater. I am happy to recommend publication so long as the following comment is addressed.

1.The contents of the "Method" column in Table 1 may need to be condensed or described in other ways because they can look cluttered. For the discussion section of FTIR, XRD and SEM, relevant pictures can be merged and reformatted appropriately. Please revise.

2. In page 18,The results of turbidity removal using chitosan from carapace of M. rosenbergii were higher than those of (Carpinteyro-Urban & Torres, 2013) and (Chen et al, 2020).”, the description should indicate the specific turbidity removal value of the references. Please revise.

3. The conclusion section should be more informative and the potential impact of the work should be highlighted. Please revise.

4. Type of references should be in consistent. For example, ref 3 is missing the DOI link. Journal names are mixed with full names and abbreviations. Please revise.

Author Response

This paper describes a simple coagulation/flocculation technique that uses aquaculture biowaste to produce chitosan coagulants for the removal of turbidity and salinity from actual aquaculture wastewater. Response Surface Methodology (RSM) was used to optimize chitosan dosage, pH, settling time and other independent parameters to obtain a higher removal efficiency. The research idea is interesting and the potential application is valuable. I believe that this paper has an important contribution to the use of RSM approach to optimize new applications of Macrobrachium rosenbergii crustaceans for the coagulation-flocculation treatment of aquaculture wastewater. I am happy to recommend publication so long as the following comment is addressed.

1.The contents of the "Method" column in Table 1 may need to be condensed or described in other ways because they can look cluttered. For the discussion section of FTIR, XRD and SEM, relevant pictures can be merged and reformatted appropriately. Please revise.

RESPONSE: Thank you for your observations and further suggestions. It can be observed that many physicochemical parameters were examined in this study. Authors were concerned about the length of the manuscript should the methods are described on separate pages. However, the methods in table 1 have been revised with the hope that the reviewer will find it appealing.

Authors also observed that it is not appropriate to merge SEM images of chitosan. We agree on the part of XRD and FTIR that they could be merged. Nevertheless, we noticed that the respective figures are only few and therefore can better be understood in detail if placed separately.

  1. In page 18, “The results of turbidity removal using chitosan from carapace of M. rosenbergiiwere higher than those of (Carpinteyro-Urban & Torres, 2013) and (Chen et al, 2020).”, the description should indicate the specific turbidity removal value of the references. Please revise.

RESPONSE: The page has been revised as instructed by the reviewer.

  1. The conclusion section should be more informative and the potential impact of the work should be highlighted. Please revise.

RESPONSE: The conclusion part of the manuscript has been revised with the hope of measuring up to the standard recommended by the reviewer.

  1. Type of references should be in consistent. For example, ref 3 is missing the DOI link. Journal names are mixed with full names and abbreviations. Please revise.

RESPONSES: The references have been revised as requested by the reviewer.

Reviewer 2 Report

This study presents a promising solution for the problem of waste generation in aquaculture operations by utilizing biowaste to produce chitosan coagulant for wastewater treatment. Through the use of chemical methods, chitin and chitosan were extracted from the carapace of Macrobrachium rosenbergii, and were then tested for their ability to remove turbidity and salinity from shrimp aquaculture wastewater. The study found that 80g of raw powder carapace yielded chitin and chitosan of 23.79% and 20.21%, respectively, with low moisture and ash content indicating good quality chitosan. Additionally, the high solubility and DDA suggested good coagulant potentials. The study also found that 87.67% turbidity was successfully removed at 20mg/L of chitosan dosage and 6.25 pH after 30 minutes settling time, and 21.43% salinity was removed at 5mg/L, 7.5- and 30-minutes chitosan dosage, pH and settling time respectively. Overall, the study provides evidence that chitosan extracted from carapace can be used as a biopolymer coagulant for aquaculture wastewater treatment. 

However, it is worth noting that the language used in the study may require some minor corrections.

Figure 1 should be reduced. Figure 2 is redundant.

The conclusions should be corrected. The presented conclusions only sum up results and don't contain information about importance and use of these studies in the future.

Author Response

This study presents a promising solution for the problem of waste generation in aquaculture operations by utilizing biowaste to produce chitosan coagulant for wastewater treatment. Through the use of chemical methods, chitin and chitosan were extracted from the carapace of Macrobrachium rosenbergii, and were then tested for their ability to remove turbidity and salinity from shrimp aquaculture wastewater. The study found that 80g of raw powder carapace yielded chitin and chitosan of 23.79% and 20.21%, respectively, with low moisture and ash content indicating good quality chitosan. Additionally, the high solubility and DDA suggested good coagulant potentials. The study also found that 87.67% turbidity was successfully removed at 20mg/L of chitosan dosage and 6.25 pH after 30 minutes settling time, and 21.43% salinity was removed at 5mg/L, 7.5- and 30-minutes chitosan dosage, pH and settling time respectively. Overall, the study provides evidence that chitosan extracted from carapace can be used as a biopolymer coagulant for aquaculture wastewater treatment.

However, it is worth noting that the language used in the study may require some minor corrections.

Figure 1 should be reduced. Figure 2 is redundant.

RESPONSE: Figure 2 is the chitosan produced in the present study. One of the physical properties of the isolated chitosan is the colour. Authors hope that the inclusion of the figure will help readers have a pictorial representation of the chitosan under consideration. Also, we believe that this figure will be of relevance to readers who may be reading about chitosan for the first time through this article.

The conclusions should be corrected. The presented conclusions only sum up results and don't contain information about importance and use of these studies in the future.

RESPONSE: The conclusion part of the manuscript has been revised.